# Trends in Treatment of Head and Neck Cancer in Germany: A Diagnosis-Related-Groups-Based Nationwide Analysis, 2005–2018

**DOI:** 10.3390/cancers13236060

**Published:** 2021-12-01

**Authors:** Isabel Hermanns, Rafat Ziadat, Peter Schlattmann, Orlando Guntinas-Lichius

**Affiliations:** 1Department of Otorhinolaryngology, Jena University Hospital, D-07747 Jena, Germany; isabelhermanns@gmail.com (I.H.); rafat.ziadat@med.uni-jena.de (R.Z.); 2Department of Medical Statistics, Computer Sciences and Data Sciences, Jena University Hospital, D-07743 Jena, Germany; Peter.Schlattmann@med.uni-jena.de

**Keywords:** head and neck neoplasm, disparity, surgery, radiation, chemotherapy/immunotherapy, nationwide

## Abstract

**Simple Summary:**

Surgery, radiotherapy, and chemotherapy/immunotherapy as monotherapy or in combination are the pillars of the treatment of head and neck cancer (HNC). Nation-wide population-based data on treatment rates per population and year for HNC are sparse. The data of virtually all HNC cases (apart from thyroid cancer) treated as inpatients in Germany between 2005 and 2018 were analyzed. Treatment rates for nearly all treatment types increased for cancer of the oral cavity, oropharynx, and salivary glands. Treatment rates for nasopharyngeal cancer in both sexes and hypopharyngeal cancer in men mainly decreased. In women, surgery for hypopharyngeal cancer decreased, but radiotherapy, chemotherapy, or in combination, increased. Laryngeal cancer showed a mixed picture: Surgery and neck dissection decreased in men and remained unchanged in women, whereas radiotherapy, chemotherapy, or in combination, remained unchanged in men, but increased in women. Changes in treatment are dependent on the subsites and are different for men and women for several subsites.

**Abstract:**

Advances in head and neck cancer (HNC) treatment might have changed treatment strategies. This study determined, with focus on gender disparity, whether treatment rates have changed for inpatients in Germany between 2005 and 2018. Nation-wide population-based diagnosis-related groups (DRG) data of virtually all HNC cases (1,226,856 procedures; 78% men) were evaluated. Poisson regression analyses were used to study changes of annual treatment rates per German population. For surgery, the highest increase was seen for women with cancer of the oral cavity (relative risk (RR) 1.14, 95% confidence interval (CI) 1.11–1.18, *p* < 0.0001) and the highest decrease for men with laryngeal cancer (RR 0.90, CI 0.87–0.93). In women with oropharyngeal cancer, the highest increase of radiotherapy rates was seen (RR 1.18, CI 1.10–1.27, *p* < 0.0001). A decrease was seen in men for hypopharyngeal cancer (RR 0.93, CI 0.87–0.98, *p* = 0.0093). The highest increase for chemotherapy/immunotherapy was seen for women with oropharyngeal cancer (RR 1.16, CI 1.08–1.24, *p* < 0.0001), and a decrease in men with hypopharyngeal cancer (RR 0.93, CI 0.88–0.97, *p* = 0.0014). Treatment patterns had changed for nearly all subsites and therapy types. There were relevant gender disparities, which cannot be explained by the DRG data.

## 1. Introduction

Head and neck cancer (HNC) is the seventh most common type of cancer worldwide [1]. HNC represents a diverse group of tumor entities covering several anatomical subsites in the upper aerodigestive region. These entities differ greatly in terms of etiology, risk factors, histology, and therapeutic management. About half of the patients are still diagnosed at advanced stage [2,3]. In most cases, a monotherapeutic approach (surgery alone or radiotherapy alone) is used for early-stage disease, whereas locally-advanced stages are treated by multimodal approaches [1,4,5]. Chemotherapy/immunotherapy regimens are mainly part of concurrent or postoperative radiochemotherapy/immunotherapy. Cetuximab as a first monoclonal antibody was licensed in 2006, first in systematic treatment concepts for recurrent or metastatic disease, and later on as substitute for chemotherapy/immunotherapy in curative radiotherapy concepts [6]. The first checkpoint inhibitor as new treatment option was licensed in 2017, hence did not play a role in the herein examined period from 2005 to 2018 [7].

Epidemiologic population-based studies on head and neck cancer are rare. Nation-wide data on head and neck cancer, i.e., complete data sets of all patients treated for head and neck cancer, are only available from some countries and are a rare exception [8,9]. Most population-based information comes from cancer registries covering only parts of the population (Europe: EUROCARE-5 study; case coverage unclear; United States: Surveillance, Epidemiology, and End Results (SEER) Program and National Cancer Database (NCDB), case coverage 30–70%) or from regional cancer registries [2,5,10,11].

In 2004, a diagnosis-related-groups (DRG)-based reimbursement system for German hospitals became statutory based on the German Hospital Reimbursement Act (KHEntgG) and the implementation of the Case Fees Act (FPG). The hospitals have to submit the DRG coding to the insurance company of the patients to receive the reimbursement of hospital stays. Furthermore, all hospitals submit their hospitalization data annually to the Hospital Remuneration System (InEK) for a continual adjustment of the DRG system. Later on, the data is anonymized and forwarded to the Federal Bureau of Statistics (DESTATIS; https://www.destatis.de/; last access: 22 September 2021). The DRG data evaluated by the Federal Statistical Office includes various socio-demographic variables, as well as extensive characteristics for diagnoses and treatment courses of all inpatients who were discharged in virtually all German hospitals during the reporting year and can be used for scientific analyses.

We used the DRG data from 2005 to 2018 to analyze nationwide treatment rates for HNC in Germany with focus on the different treatment options, the influence of gender, and trends over the years.

## 2. Material and Methods

All hospitalizations for the years 2005–2018 in this study covering virtually all hospitals in Germany were analyzed. The structure of DRG data supplied by the Federal Bureau of Statistics (DESTATIS) has been described elsewhere in detail [12] (more information about the data source is available on https://www.forschungsdatenzentrum.de/en/health/drg; last access: 22 September 2021). Ethics approval was not needed. The authors used anonymized data supplied by the German Federal Bureau of Statistics (DESTATIS). The anonymization of such data is regulated in § 16 Bundesstatistikgesetz (German Federal Statistics Act). All authors had access to the study data and reviewed and approved this study.

### 2.1. Patient Cohort Definition

The hospitalizations with a primary or secondary diagnosis for head and neck cancer of the International Classification of Diseases, 10th Edition, German Modification (ICD-10-GM) were analyzed: C00, C01-C06, C07-C14, C30-C33, and C77.0. Patients with thyroid cancer (C73) were not included. Subsequently, all cases were grouped according to the OPS procedures (Operationen- und Prozedurenschlüssel; OPS, version 2005 to 2018): 1-24 = examination in the head and neck area; 1-41 = biopsy without incision of the eye, ear, nose and skin of the face, head and neck; 1-42 = biopsy without incision of the mouth, oral cavity, larynx, pharynx, and blood-forming organs; 1-43 = biopsy without incision of respiratory organs; 1-53 = biopsy through incision on the ear and nose; 1-54 = biopsy through incision of the mouth, oral cavity and pharynx; 5-21 = nasal surgery; 5-22 = paranasal sinus surgery; 5-25 = tongue surgery; 5-26 = salivary glands and salivary gland duct surgery; 5-27 = other mouth and face surgery; 5-28 = surgery in the area of the nasopharynx and oropharynx; 5-29 = pharyngeal surgery; 5-30 = excision and resection of the larynx; 5-301 = hemilaryngectomy; 5-31 = other laryngeal and tracheal surgeries; 5-401.0 = excision of individual cervical lymph nodes and lymph vessels, including removal of several sentinel lymph nodes; 5-403 = radical cervical lymphadenectomy (neck dissection); 8-52 = radiation therapy; 8-54 = chemotherapy, immunotherapy, and antiretroviral therapy (hereinafter: chemotherapy/immunotherapy). In addition, it was in addition possible to analyze of the large entities (oral cavity C01-C06, oropharynx C10, hypopharynx C12-C13, larynx C32) according to the following OPS-codes in more detail: 5-270 = maxillofacial incision and drainage; 5-271 = incision of hard and soft palate; 5-272 = excision and destruction of diseased hard and soft palate; 5-273 = incision, excision and destruction in the oral cavity; 5-274 = floor of the mouth reconstruction; 5-275 = palatoplasty; 5-277 = resection of the floor of the mouth and reconstruction; 5-278 = resection of the cheek and reconstruction; 5-290 = pharyngotomy; 5-292 = excision and destruction of diseased pharyngeal tissue; 5-293 = pharyngoplasty; 5-294 = other reconstruction of the pharynx; 5-295 = partial resection of the pharynx; 5-296 = radical resection of the pharynx; 5-300 = excision and destruction of diseased tissue of the larynx; 5-301 = hemilaryngectomy; 5-302 = other partial laryngectomy; 5-303 = laryngectomy; 5-401 = excision of neck lymph nodes and lymph vessels; and 5-403 = radical neck dissection. Finally, the variable gender (male, female) was analyzed within the extracted data.

### 2.2. Statistical Analysis

The population of Germany of the years 2005–2018 was used to calculate the treatment rates. Population data were provided by the German Federal Bureau of Statistics (DESTATIS). Negative binomial regression models with log link were performed to conduct an analysis over time. Here, the dependent variable was the number of cases and the logarithm of the population at risk was taken as an offset. Time from 2005 was used as covariate. Relative risks (RR) with 95% confidence intervals (CI) are reported. For all statistical tests, significance was two-sided and set to *p* < 0.05. All calculations were carried out using SAS version 9.4 (SAS Institute, Cary, NC, USA). The changes of the treatment rates between the first year, 2005, and the last year, 2018, were calculated as follows: ((treatment rate 2018 − treatment rate 2005)/treatment rate 2005) × 100.

## 3. Results

### 3.1. Treatment Strategies and Average Treatment Rates

Overall, 1,226,856 procedures (78.2% men) performed as in-patient treatment for patients with HNC in German hospitals between 2005 and 2018 were analyzed. This included 217,859 biopsies (average per year: 18,154.92 ± 16,560.20), 378,151 surgeries of primary HNC (average: 31,512.58 ± 32,480.92), 152,207 neck dissections (average: 12,683.92 ± 11,855.29), 237,728 radiotherapy treatments (average: 19,810.67 ± 15,950.13), and 240,911 chemotherapy/immunotherapy treatments (average: 20,075.92 ± 16,353.38). The average treatment rates per 100,000 population per year are shown in Table 1. Appendix A illustrates the changes of the treatment rates over time for the different types of treatments and separately for the head and neck cancers subsites. Overall, the highest average biopsy rate was seen for laryngeal cancer (5.32 ± 0.70 per 100,000 per year) and cancer of the oral cavity (4.88 ± 0.79). The same was seen for the frequency of surgery of the primary tumors with a surgery rate of 10.96 ± 0.48 for laryngeal cancer and of 10.25 ± 0.74 for cancer of the oral cavity. The neck dissection rates were dominated by neck dissection for cancer of the oral cavity (5.34 ± 0.41). In general, the neck dissection rates were much lower than the primary surgery rates. Radiotherapy rates were headed by cancer of the oral cavity (5.95 ± 0.54), followed by oropharyngeal cancer (4.13 ± 0.44). The chemotherapy/immunotherapy rates were dominated by oral cavity cancer (5.82 ± 0.59), oropharyngeal cancer (4.15 ± 0.42), and hypopharyngeal cancer (4.08 ± 0.39). In general, the treatment rates were much higher for men than for women. The treatment rates over time are shown for both sexes separately in Figure 1. The frequency ranking was different between both sexes for the biopsy and primary surgery rates: Laryngeal cancer led biopsies and primary surgery in men and oral cavity was in second place, whereas it was vice versa in women. The ranking of the frequencies between the subsites was not different for both sexes concerning neck dissection, radiotherapy, and chemotherapy/immunotherapy.

### 3.2. Gender Differences in Change of Therapy Strategies between 2005 and 2018

Table 2 summarizes the treatment rates for each subsite for both sexes, and based on the regression analyses, if these rates have increased, decreased, or were unchanged over the years. Furthermore, the changes of treatment rates in percentage between the first year, 2005, and the last year, 2018, are listed.

The biopsy rates increased over time for all subsites (RR 1.09–1.37; all *p* < 0.0001), but less noticeable for hypopharyngeal cancer in men (RR 1.07; *p* < 0.05; Table 3).

The rates of surgery of the primary tumor significantly increased only in women for oral cavity cancer (RR 1.14, CI 1.11–1.18, *p* < 0.0001), oropharyngeal cancer (RR 1.04, CI 1.00–1.07, *p* = 0.0281), nasopharyngeal cancer (RR 1.06, CI 1.03–1.10, *p* = 0.0006), and for both sexes for salivary gland cancer (men: RR 1.07, CI 1.03–1.11, *p* = 0.0001; women: RR 1.07, CI 1.05–1.09, *p* < 0.0001; Table 4). In contrast, surgery of the hypopharynx decreased for both sexes (men: RR 0.91, CI 0.88–0.95, *p* < 0.0001; women: RR 0.92, CI 0.86–0.98, *p* = 0.0071). For men, surgery rates decreased for laryngeal cancer (RR 0.90., CI 0.87–0.93, *p* < 0.0001), and for oropharyngeal cancer (RR 0.95, CI 0.93–0.97, *p* < 0.0001).

The neck dissection rates increased for both sexes over the years for cancer of the oral cavity (men: RR 1.04, CI 1.01–1.07, *p* = 0.0087; women: RR 1.19, 1.15–1.22, *p* < 0.0001) and salivary glands (men: RR 1.08, CI 1.04–1.12, *p* = 0.0002; women: RR 1.07, CI 1.04–1.10, *p* < 0.0001), but decreased for cancer of the nasopharynx (men: RR 0.86, CI 0.80–0.92, *p* < 0.0001; women: RR 0.86, CI 0.76–0.97, *p* = 0.0120) and hypopharynx (RR 0.85, 0.81–0.89, *p* < 0.0001; women: RR 0.91, CI 0.85–0.97, *p* = 0.0030, Table 5). Moreover, the neck dissection rate decreased for laryngeal cancer in men (RR 0.89, CI 0.87–0.90, *p* < 0.0001). The neck dissection rates for oropharyngeal cancer showed no changing trend.

Radiotherapy showed increasing rates for both sexes for cancer of the oral cavity (men: RR 1.08, CI 1.04–1.12, *p* = 0.0002; women: RR 1.12, CI 1.07–1.18, *p* < 0.0001; Table 6), and salivary glands (men: RR 1.09, CI 1.04–1.14, *p* = 0.0003; women: RR 1.12, CI 1.06–1.18. *p* < 0.0001). The radiotherapy rate also increased for women with oropharyngeal cancer (RR 1.18, CI 1.10–1.27, *p* < 0.0001) and with laryngeal cancer (RR 1.17, CI 1.09–1.24, *p* < 0.0001). The radiotherapy rate decreased for nasopharyngeal cancer in men and women (men: RR 0.93, CI 0.88–0.98, *p* = 0.0085; trend for women: RR 0.94, CI 0.87–1.01, *p* = 0.0748), and in men for hypopharyngeal cancer (RR 0.92, CI 0.87–0.98; *p* = 0.0093).

Furthermore, for chemotherapy/immunotherapy, some subsites showed an increase and others a decrease of treatment rates (Table 7). The treatment rates for cancer of the oral cavity (men: RR 1.09, CI 1.04–1.14, *p* = 0.0005; women: RR 1.13, CI 1.06–1.20, *p* < 0.0001), salivary glands (men: RR 1.10, CI 1.03–1.17, *p* = 0.0049; women: RR 1.12, CI 1.05–1.21, *p* = 0.0015), and oropharynx (men: RR 1.05, 0.99–1.12, *p* = 0.0742; women: RR 1.16, CI 1.08–1.24, *p* < 0.0001) increased for both sexes. For laryngeal cancer, the rate increased only for women (RR 1.07, CI 1.01–1.13, *p* = 0.0337). In contrast, a decrease of the chemotherapy/immunotherapy rate was seen for men with nasopharyngeal cancer (RR 0.95, CI 0.91–0.99, *p* = 0.0193) and hypopharyngeal cancer (RR 0.93, CI 0.88–0.97, *p* = 0.0014).

For the major subsites, a more detailed analysis of the most frequent procedures was feasible. Figure 2 shows the treatment rates over the time from 2005 to 2018 separately for cancer of the oral cavity, oropharynx, hypopharynx, and larynx. For cancer of the oral cavity, conservative treatment increased over the years (radiotherapy: RR 1.09, CI 1.05–1.13, *p* < 0.0001; chemotherapy/immunotherapy: RR 1.10, CI 1.05–1.15, *p* = 0.0001; Appendix A). Cancer surgery of the palate (RR 1.07, CI 1.02–1.11, *p* = 0.0017), cheek (RR 1.23, CI 1.16–1.30, *p* < 0.0001), and coding of radical neck dissection (RR 1.09; CI 1.06–1.12, *p* < 0.0001) increased. The only significant decrease was seen for surgery of the floor of the mouth (RR 0.90, CI 0.84–0.97, *p* = 0.0032). For oropharyngeal cancer, radiotherapy (RR 1.08, CI 1.02–1.14, *p* = 0.0064) and chemotherapy/immunotherapy (RR 1.08, CI 1.02–1.14, *p* = 0.0113) increased (Appendix A). Concerning surgical procedures, partial resection of the pharynx (RR 1.078, CI 1.03–1.13, *p* = 0.0007), and radical resection of the pharynx (RR 1.33, CI 1.15–1.54, *p* = 0.0001) increased. Pharyngotomy (RR 0.79, CI 0.73–0.86, *p* < 0.0001), excision of pharyngeal tissue (RR 0.75, CI 0.67–0.85, *p* < 0.0001), and pharyngoplasty (RR 0.67, CI 0.60–0.76, *p* < 0.0001) decreased. Interestingly, the treatment rates for the most important procedures all decreased for hypopharyngeal cancer, apart from radical resection of the pharynx (RR 1.18, CI 1.05–1.33, *p* = 0.0051, Appendix A). Radiotherapy (RR 0.94, CI 0.88–0.99, *p* = 0.0323) and chemotherapy/immunotherapy rates (RR 0.93, CI 0.89–0.98, *p* = 0.0089) decreased as a treatment of hypopharyngeal cancer. Pharyngotomy (RR 0.64, CI 0.59–0.69, *p* < 0.0001), excision diseased pharyngeal tissue (RR 0.74, CI 0.68–0.81, *p* < 0.0001), pharyngoplasty (RR 0.61, CI 0.53–0.72, *p* < 0.0001), and partial resection of the pharynx (RR 0.89, CI 0.81–0.963, *p* = 0.0051) as treatment options all decreased. The procedures including surgery of the larynx, i.e., excision of laryngeal tissue (RR 0.75, CI 0.71–0.79, *p* < 0.0001), hemilaryngectomy (RR 0.72, CI 0.64–0.81, *p* < 0.0001), other partial laryngectomy (RR 0.73, CI 0.69–0.78, *p* < 0.0001), and laryngectomy (RR 0.94, CI 0.90–0.98, *p* = 0.0075) also decreased. The rate of excision of neck lymph nodes (RR 0.87, CI 0.78–0.96, *p* = 0.0080) as well as the radical neck dissection rate (RR 0.86, CI 0.82–0.90, *p* < 0.0001) also decreased. Finally, conservative treatment of laryngeal cancer was unchanged from 2005 to 2018 (Appendix A). This was true for radiotherapy (RR 1.03, CI 0.99–1.08, *p* = 0.1779) and chemotherapy/immunotherapy (RR 1.02, CI 0.98–1.05, *p* = 0.4304). The most important surgical procedures rates all decreased: excision of laryngeal tissue (RR 0.89, CI 0.87–0.91, *p* < 0.0001), hemilaryngectomy (RR 0.79, CI 0.74–0.83, *p* < 0.0001), other partial laryngectomy (RR 0.95, CI 0.93–0.98, *p* < 0.0001), laryngectomy (RR 0.95, CI 0.93–0.97, *p* < 0.0001), excision of neck lymph nodes (RR 0.84, CI 0.77–0.92, *p* = 0.0002), and radical neck dissection (RR 0.91, CI 0.89–0.92, *p* < 0.0001).

## 4. Discussion

Overall, the burden of cancer incidence including HNC is rapidly growing worldwide. This reflects both aging and growth of the population as well as changes in the prevalence and distribution of the main risk factors for cancer [13]. Increasing numbers of HNC cases will need increasing numbers of treatment. The presented study is based on secondary data analysis from DRG data. Several countries use the DRG system, but to our knowledge the present study is the first to use DRG data to analyze head and neck treatment trends for over time. This data set, incorporating as a major advantage all hospitalized cases in Germany, is based on reimbursement claims. Hence, and this a limitation of the study, it does not include outpatient treatment [14]. Therefore, the absolute number of treatments for HNC will be higher than presented here. The comprehensive picture of inpatient treatment relies on OPS treatment codes that can be linked to ICD codes. The herein presented treatment rates per 100,000 population and year cannot be transferred par for par to incidence rates as some HNC patients, especially when in advanced stage, might receive several treatments.

As no other DRG-based analysis was performed before, the presented results could only be compared to publications based on other population-based data sources, i.e., mainly on cancer registry data. Like in many other countries, it is the task of population-based cancer registries in Germany to estimate and analyze the number of annual incident cancer cases, cancer deaths, survival rates, and additional indicators of cancer epidemiology, particularly prevalence, incidence, and including the investigation of trends over time. Additionally, it is the task of the clinical cancer registries to collect further clinical data, for instance the TNM staging. Therefore, the presented analysis did not allow a linkage of the presented treatment data to TNM staging or any other clinical data. Due to the German Centre for Cancer Registry Data (https://www.krebsdaten.de; accessed on 22 October 2021), for instance, the average incidence of cancer of the oral cavity between 2005 and 2017 (data of 2018 not yet available) was 24.8 for men and 9.6 for women, respectively. The average incidence of laryngeal cancer between 2005 and 2017 was 8.3 for men and 1.3 for women, respectively. In comparison, the DRG data analyses revealed for cancer of the oral cavity a cumulative treatment rate of 46.6 for men and 18.4 for women, respectively. The cumulative treatment rates for laryngeal cancer were 25.6 and 6.8, respectively. This reflects that most patients receive several procedures and not monotherapy. It is known that the hospitalization rates range between 1 and 2 fold of the incidence rates [15].

An older National Cancer Data Base (NCDB) analysis for the years 1990–2004 showed decreasing numbers of surgery, a decrease of radiotherapy as monotherapy, but a massive increase of radiochemotherapy and a slight increase of surgery combined with radiochemotherapy [16]. Cooper et al. relate these changes mainly to the emergence of data from phase III protocols showing that the addition of chemotherapy to radiation therapy enhances the rate of locoregional control in advanced HNC tumors. A Surveillance, Epidemiology, and End Results (SEER) analysis for the years 1997, 2004, and 2009 showed mainly the same results [17]. We are not aware of newer NCDB or SEER analyses covering actual years with focus on treatment trends. More appropriate is a population-based analysis using clinical cancer registry data from Thuringia, a federal state in Germany, covering the years 1996–2016 [5]. Here, surgery alone (26.5% of all cases), surgery with adjuvant radiochemotherapy (21.2%), surgery with adjuvant radiotherapy (21.0%), and definitive radiochemotherapy/radioimmunotherapy (11.9%) were in descending order the most predominant strategies. Furthermore, the relative frequency of radiotherapy as a single therapy decreased, instead, radiochemotherapy/radio-immunotherapy increased. In contrast to the mentioned NCDB and SEER data, surgery as a single modality, as a primary treatment in combination with radiochemotherapy/radio-immunotherapy increased in Germany. It seems that surgery plays a larger role even for advanced stage tumors in Germany than compared to the United States. Here, we could separately analyze the surgery of the primary and surgery of the neck. It is striking that the neck dissection rates for all localizations were >50% lower than the surgery rates for the primary tumor. This means that an elective neck dissection was probably not performed in many early cancer cases (stage I) with clinical N0. Whereas this is standard of care or at least matter of debate for most head neck subsites, neck dissection is standard for cancer of the oral cavity even for all N0 necks and independent of the T classification. This suggests that part of the cases with cancer of the oral cavity were undertreated. The neck dissection rate is a quality indicator for the treatment of cancer of the oral cavity for the certified head and neck cancer centers in Germany. In 2020, only 76% of the oral cavity cancer cases treated in certified centers received a neck dissection (https://www.onkozert.de/organ/kopf-hals/; last access: 22 September 2021). Even taking into account that some patients refused a neck dissection or it was not indicated because of multimorbidity, an undertreatment can be assumed. This shows that more effort has to be made to improve the clinical guideline adherence.

Finally, the use of chemotherapy/biologicals as part of the treatment concept increased. The increase of biologicals in mainly related to cetuximab introduced in 2006 as checkpoint inhibitors were not introduced before 2017 [6,7]. These German cancer registry data correspond better to the presented results, especially as factors, such as the subsite and gender, have to be considered in the presented SEER and NCDB data. Of main interest are inverse trends. For most therapy types, we revealed increasing treatment rates over time, but especially treatment rates for nasopharyngeal cancer and hypopharyngeal cancer, as well as laryngeal cancer in male patients decreased. This seems to be directly associated to a decreasing incidence of these tumor subtypes in Germany [5].

Although it is well known that females have improved survival in comparison with their male counterparts [18], gender disparities were especially neglected so far when analyzing treatment trends. The above-mentioned Thuringian data showed that treatment decisions were different between male and female patients even for the same tumor type and stage [5]. Understanding these gender disparities is essential to providing appropriate care to HNC patients [18]. Therefore, prospective trials are needed to better analyze the decision making for or against a certain treatment strategy.

Although DRG data is collected prospectively through the routine hospital coding process which collects data from virtually all German hospitals, this study has several limitations. First, this study is retrospective, which could lead to misclassification errors and unmeasured variables. Notably, many clinical but important factors with influence on decision making, such as stage, comorbidity, or age, were not included. The calculated treatment rates can only be seen as a proxy for the incidence rates [15]. Furthermore, the DRG data is primarily collected for reimbursement. Although the data underwent plausibility checks for incorrect codes before release of DESTATIS to researchers, it cannot be excluded that in some cases the coding followed the interest of maximizing the profit more than a proper documentation of the actual treatment [19]. As the complete dataset is subject to this uncontrolled bias, the comparisons between subsets, such as the comparisons between men and women or between subsites, are evenly affected by such a bias.

## 5. Conclusions

The German and nationwide DRG statistics provide a unique epidemiological data source for the quantification of treatment rates for HNC. For the first time representative population-based DRG data of 1,226,856 procedures performed between 2005 and 2018 were used to analyze treatment trends in HNC in Germany. HNC is a very heterogeneous type of cancer. Therapy decisions are very different between HNC subtypes and trends over time are dependent on the subtype. Moreover, treatment decision also seems to depend on gender. These differences in decision making between men and women have to be better understood.

## Figures and Tables

**Figure 1 cancers-13-06060-f001:**
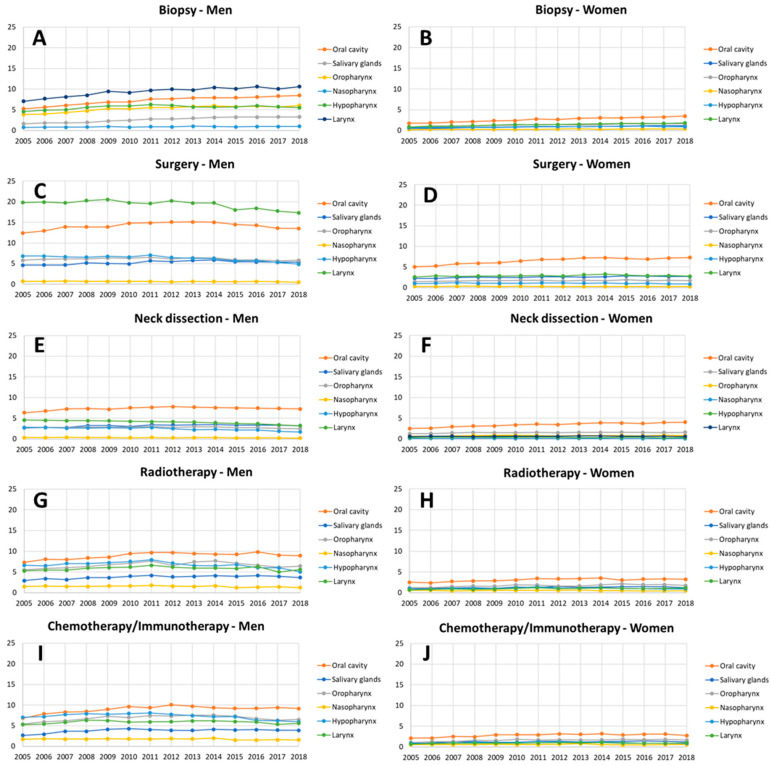
Annual treatment rates per 100,000 population for head and neck cancer from 2005 to 2018, separately for men (**A**,**C**,**E**,**G**,**I**) and women (**B**,**D**,**F**,**H**,**J**). (**A**,**B**) Biopsy rates; (**C**,**D**) surgery of the primary tumor rates; (**E**,**F**) neck dissection rates; (**G**,**H**) radiotherapy rates, and (**I**,**J**) chemotherapy/immunotherapy rates.

**Figure 2 cancers-13-06060-f002:**
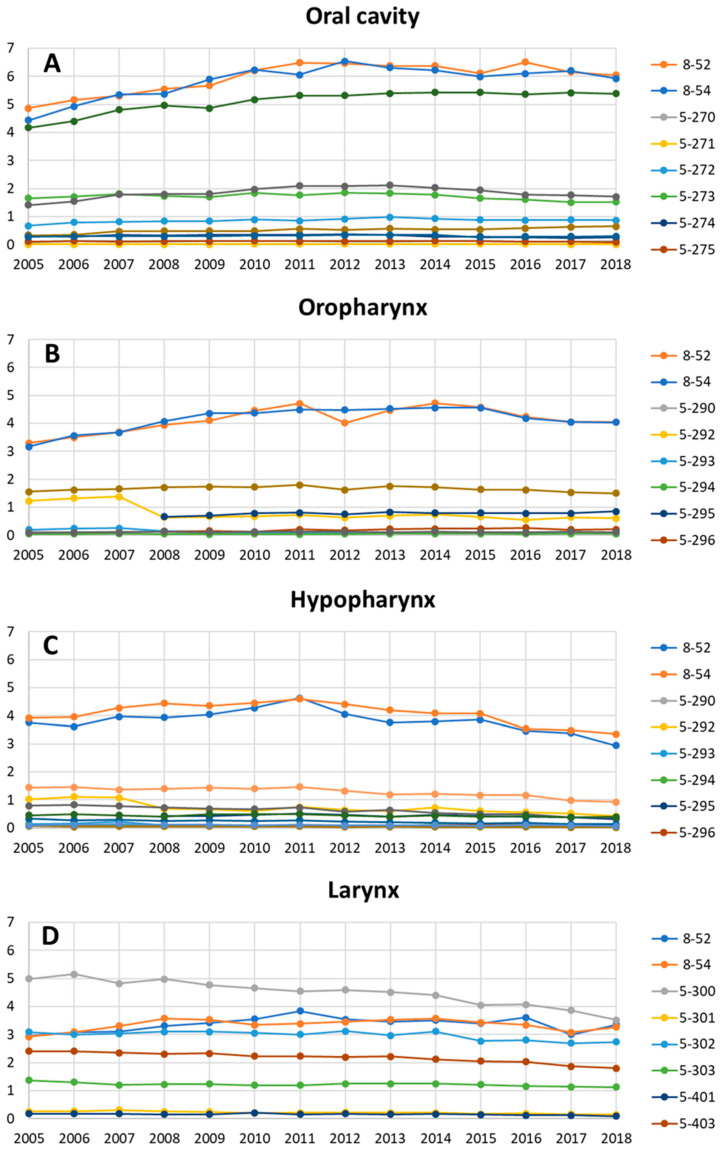
Annual operation and procedure rates per 100,000 population for different subsites of head and neck cancer. (**A**) Oral cavity; (**B**) oropharynx; (**C**) hypopharynx; (**D**) larynx. 8-52 = radiotherapy; 8-54 = chemotherapy/immunotherapy; 5-270 = maxillofacial incision and drainage; 5-271 = incision of hard and soft palate; 5-272 = excision and destruction of diseased hard and soft palate; 5-273 = incision, excision and destruction in the oral cavity; 5-274 = floor of the mouth reconstruction; 5-275 = palatoplasty; 5-277 = resection of the floor of the mouth and reconstruction; 5-278 = resection of the cheek and reconstruction; 5-290 = pharyngotomy; 5-292 = excision and destruction of diseased pharyngeal tissue; 5-293 = pharyngoplasty; 5-294 = other reconstruction of the pharynx; 5-295 = partial resection of the pharynx; 5-296 = radical resection of the pharynx; 5-300 = excision and destruction of diseased tissue of the larynx; 5-301 = hemilaryngectomy; 5-302 = other partial laryngectomy; 5-303 = laryngectomy; 5-401 = excision of neck lymph nodes and lymph vessels; 5-403 = radical neck dissection.

**Table 1 cancers-13-06060-t001:** Average treatment rates per 100,000 population per year of different types of therapy for head and neck cancer in Germany, 2005–2018.

Procedure/Localization	All	Male	Female
	Mean ± SD	Mean ± SD	Mean ± SD
Biopsy			
All localizations	3.17 ± 1.73	5.17 ± 2.91	1.24 ± 0.78
Oral cavity	4.88 ± 0.79	7.22 ± 1.03	2.63 ± 0.56
Salivary glands	1.74 ± 0.42	2.63 ± 0.61	0.88 ± 0.23
Oropharynx	3.26 ± 0.5	5.27 ± 0.74	1.32 ± 0.28
Nasopharynx	0.60 ± 0.06	0.90 ± 0.08	0.31 ± 0.05
Hypopharynx	3.21 ± 0.29	5.61 ± 0.48	0.89 ± 0.11
Larynx	5.32 ± 0.7	9.38 ± 1.12	1.41 ± 0.29
Surgery			
All localizations	5.50 ± 3.84	8.63 ± 6.28	2.49 ± 2.04
Oral cavity	10.25 ± 0.74	14.14 ± 0.83	6.51 ± 0.77
Salivary glands	3.88 ± 0.27	5.25 ± 0.42	2.56 ± 0.17
Oropharynx	3.85 ± 0.15	6.11 ± 0.26	1.68 ± 0.12
Nasopharynx	0.46 ± 0.04	0.66 ± 0.07	0.26 ± 0.03
Hypopharynx	3.61 ± 0.34	6.27 ± 0.64	1.04 ± 0.09
Larynx	10.96 ± 0.48	19.33 ± 1.02	2.88 ± 0.17
Neck dissection			
All localizations	2.21 ± 1.59	3.34 ± 2.15	1.13 ± 1.14
Oral cavity	5.34 ± 0.41	7.32 ± 0.39	3.42 ± 0.49
Salivary glands	2.32 ± 0.18	3.18 ± 0.26	1.50 ± 0.11
Oropharynx	1.76 ± 0.09	2.78 ± 0.16	0.78 ± 0.06
Nasopharynx	0.19 ± 0.04	0.29 ± 0.05	0.10 ± 0.02
Hypopharynx	1.37 ± 0.19	2.41 ± 0.34	0.37 ± 0.05
Larynx	2.26 ± 0.22	4.05 ± 0.42	0.63 ± 0.04
Radiotherapy			
All localizations	3.46 ± 1.56	5.58 ± 2.45	1.42 ± 0.84
Oral cavity	5.95 ± 0.54	8.95 ± 0.75	3.06 ± 0.36
Salivary glands	2.49 ± 0.25	3.75 ± 0.37	1.26 ± 0.15
Oropharynx	4.13 ± 0.44	6.68 ± 0.67	1.68 ± 0.29
Nasopharynx	1.01 ± 0.1	1.51 ± 0.16	0.52 ± 0.06
Hypopharynx	3.82 ± 0.41	6.72 ± 0.73	1.02 ± 0.15
Larynx	3.37 ± 0.26	5.85 ± 0.44	0.97 ± 0.15
Chemotherapy/Immunotherapy			
All localizations	3.51 ± 1.51	5.74 ± 2.45	1.36 ± 0.74
Oral cavity	5.82 ± 0.59	8.96 ± 0.85	2.80 ± 0.36
Salivary glands	2.49 ± 0.3	3.78 ± 0.45	1.23 ± 0.17
Oropharynx	4.15 ± 0.42	6.80 ± 0.67	1.59 ± 0.24
Nasopharynx	1.15 ± 0.09	1.74 ± 0.14	0.58 ± 0.08
Hypopharynx	4.08 ± 0.39	7.26 ± 0.69	1.03 ± 0.14
Larynx	3.34 ± 0.2	5.86 ± 0.34	0.91 ± 0.09
Sum of all procedures			
All localizations	18.04 ± 15.12	28.45 ± 14.56	7.64 ± 13.83
Oral cavity	32.51 ± 2.46	46.59 ± 3.56	18.42 ± 2.46
Salivary glands	13.02 ± 0.77	18.6 ± 1.92	7.44 ± 0.77
Oropharynx	17.35 ± 0.93	27.64 ± 2.15	7.06 ± 0.93
Nasopharynx	3.44 ± 0.15	5.10 ± 0.35	1.78 ± 0.15
Hypopharynx	16.32 ± 0.42	28.28 ± 2.37	4.35 ± 0.42
Larynx	25.64 ± 0.64	44.49 ± 1.77	6.80 ± 0.64

SD = standard deviation.

**Table 2 cancers-13-06060-t002:** Summary of the changes * of the treatment rates between 2005 and 2018.

Procedure/Localization	Results of the Regression Analyses		Changes between the First Year 2005 and the Last Year 2018	
	Male	Female	Male	Female
Biopsy				
Oral cavity	+	+	+61%	+99%
Salivary glands	+	+	+105%	+124%
Oropharynx	+	+	+56%	+94%
Nasopharynx	+	+	+26%	+55%
Hypopharynx	+	+	+21%	+35%
Larynx	+	+	+50%	+115%
Surgery				
Oral cavity	Ø	+	+9%	+44%
Salivary glands	+	+	+15%	+19%
Oropharynx	-	+	0%	+22%
Nasopharynx	Ø	+	−30%	−9%
Hypopharynx	-	-	−29%	−9%
Larynx	-	Ø	−13%	+8%
Neck dissection				
Oral cavity	+	+	+15%	+60%
Salivary glands	+	+	+6%	+29%
Oropharynx	Ø	Ø	−6%	+15%
Nasopharynx	-	-	−29%	−19%
Hypopharynx	-	-	−38%	−15%
Larynx	-	Ø	−30%	+1%
Radiotherapy				
Oral cavity	+	+	+22%	+29%
Salivary glands	+	+	+25%	+16%
Oropharynx	+	+	+16%	+50%
Nasopharynx	-	Ø	−19%	−17%
Hypopharynx	-	Ø	−24%	−11%
Larynx	Ø	+	+6%	+69%
Chemotherapy/Immunotherapy				
Oral cavity	+	+	+34%	+30%
Salivary glands	+	+	+44%	+42%
Oropharynx	Ø	+	+20%	+57%
Nasopharynx	-	Ø	−9%	+4%
Hypopharynx	-	Ø	−16%	−12%
Larynx	Ø	+	+6%	+44%

* Change (increase or decrease) only indicated if *p* < 0.05; + = increase; - = decrease; Ø = unchanged.

**Table 3 cancers-13-06060-t003:** Change of head and neck cancer biopsy rates in Germany over the time from 2005 to 2018 for both sexes.

Localization	Sex	Estimate	StdErr	ChiSq	*p*	RR	(95% CI)
Oral cavity	m	0.0340	0.0031	122.31	<0.0001	1.19	1.15–1.22
Oral cavity	f	0.0515	0.0030	297.05	<0.0001	1.29	1.26–1.33
Salivary glands	m	0.0574	0.0047	149.47	<0.0001	1.33	1.27–1.4
Salivary glands	f	0.0637	0.0045	204.17	<0.0001	1.38	1.32–1.44
Oropharynx	m	0.0321	0.0042	57.09	<0.0001	1.17	1.13–1.22
Oropharynx	f	0.0510	0.0050	103.81	<0.0001	1.29	1.23–1.36
Nasopharynx	m	0.0169	0.0036	21.94	<0.0001	1.09	1.05–1.13
Nasopharynx	f	0.0290	0.0066	19.49	<0.0001	1.16	1.08–1.23
Hypopharynx	m	0.0120	0.0048	6.32	0.0119	1.06	1.01–1.11
Hypopharynx	f	0.0238	0.0049	23.13	<0.0001	1.13	1.07–1.18
Larynx	m	0.0273	0.0034	65.06	<0.0001	1.15	1.11–1.18
Larynx	f	0.0487	0.0041	141.36	<0.0001	1.28	1.23–1.33

m = male; f = female; StdErr = standard error; ChiSq = chi-square distribution; CI = confidence interval.

**Table 4 cancers-13-06060-t004:** Change of surgery rates of the primary head and neck cancer in Germany over the time from 2005 to 2018 for both sexes.

Localization	Sex	Estimate	StdErr	ChiSq	*p*	RR	(95% CI)
Oral cavity	m	0.0057	0.0035	2.63	0.1047	1.03	0.99–1.07
Oral cavity	f	0.0268	0.0033	65.93	<0.0001	1.14	1.11–1.18
Salivary glands	m	0.0138	0.0036	14.40	0.0001	1.07	1.03–1.11
Salivary glands	f	0.0138	0.0021	41.83	<0.0001	1.07	1.05–1.09
Oropharynx	m	−0.0101	0.0021	22.13	<0.0001	0.95	0.93–0.97
Oropharynx	f	0.0073	0.0033	4.82	0.0281	1.04	1.00–1.07
Nasopharynx	m	−0.0029	0.0027	1.17	0.2801	0.99	0.96–1.01
Nasopharynx	f	0.0118	0.0034	11.78	0.0006	1.06	1.03–1.10
Hypopharynx	m	−0.0180	0.0043	17.98	<0.0001	0.91	0.88–0.95
Hypopharynx	f	−0.0170	0.0063	7.26	0.0071	0.92	0.86–0.98
Larynx	m	−0.0219	0.0036	37.57	<0.0001	0.90	0.87–0.93
Larynx	f	−0.0076	0.0052	2.08	0.1492	0.96	0.91–1.01

m = male; f = female; StdErr = standard error; ChiSq = chi-square distribution; CI = confidence interval.

**Table 5 cancers-13-06060-t005:** Change of neck dissection rates for head and neck cancer in Germany over the time from 2005 to 2018 for both sexes.

Localization	Sex	Estimate	StdErr	ChiSq	*p*	RR	(95% CI)
Oral cavity	m	0.0076	0.0029	6.88	0.0087	1.04	1.01–1.07
Oral cavity	f	0.0342	0.0030	133.24	<0.0001	1.19	1.15–1.22
Salivary glands	m	0.0145	0.0039	14.21	0.0002	1.08	1.04–1.12
Salivary glands	f	0.0134	0.0031	18.50	<0.0001	1.07	1.04–1.10
Oropharynx	m	−0.0055	0.0036	2.40	0.1216	0.97	0.94–1.01
Oropharynx	f	0.0075	0.0050	2.29	0.1306	1.04	0.99–1.09
Nasopharynx	m	−0.0311	0.0073	17.89	<0.0001	0.86	0.80–0.92
Nasopharynx	f	−0.0306	0.0122	6.32	0.0120	0.86	0.76–0.97
Hypopharynx	m	−0.0321	0.0044	52.16	<0.0001	0.85	0.82–0.89
Hypopharynx	f	−0.0193	0.0065	8.79	0.0030	0.91	0.85–0.97
Larynx	m	−0.0243	0.0021	134.42	<0.0001	0.89	0.87–0.90
Larynx	f	0.0008	0.0043	0.04	0.8466	1.00	0.96–1.05

m = male; f = female; StdErr = standard error; ChiSq = chi-square distribution; CI = confidence interval.

**Table 6 cancers-13-06060-t006:** Change of radiotherapy rates for head and neck cancer in Germany over the time from 2005 to 2018 for both sexes.

Localization	Sex	Estimate	StdErr	ChiSq	*p*	RR	(95% CI)
Oral cavity	m	0.0150	0.0040	14.24	0.0002	1.08	1.04–1.12
Oral cavity	f	0.0232	0.0050	21.13	<0.0001	1.12	1.07–1.18
Salivary glands	m	0.0173	0.0048	13.15	0.0003	1.09	1.04–1.14
Salivary glands	f	0.0225	0.0052	19.04	<0.0001	1.12	1.06–1.18
Oropharynx	m	0.0104	0.0060	2.98	0.0846	1.05	0.99–1.12
Oropharynx	f	0.0335	0.0073	21.00	<0.0001	1.18	1.10–1.27
Nasopharynx	m	−0.0152	0.0058	6.93	0.0085	0.93	0.88–0.98
Nasopharynx	f	−0.0127	0.0072	3.17	0.0748	0.94	0.87–1.01
Hypopharynx	m	−0.0157	0.0060	6.76	0.0093	0.92	0.87–0.98
Hypopharynx	f	−0.0014	0.0097	0.02	0.8865	0.99	0.90–1.09
Larynx	m	0.0016	0.0049	0.11	0.7410	1.01	0.96–1.06
Larynx	f	0.0306	0.0066	21.51	<0.0001	1.17	1.09–1.24

m = male; f = female; StdErr = standard error; ChiSq = chi-square distribution; CI = confidence interval.

**Table 7 cancers-13-06060-t007:** Change of chemotherapy/immunotherapy rates for head and neck cancer in Germany over the time from 2005 to 2018 for both sexes.

Localization	Sex	Estimate	StdErr	ChiSq	*p*	RR	(95% CI)
Oral cavity	m	0.0165	0.0047	12.10	0.0005	1.09	1.04–1.14
Oral cavity	f	0.0244	0.0061	15.95	<0.0001	1.13	1.06–1.20
Salivary glands	m	0.0189	0.0067	7.93	0.0049	1.10	1.03–1.17
Salivary glands	f	0.0231	0.0073	10.10	0.0015	1.12	1.05–1.21
Oropharynx	m	0.0107	0.0060	3.19	0.0742	1.05	0.99–1.12
Oropharynx	f	0.0292	0.0069	17.76	<0.0001	1.16	1.08–1.24
Nasopharynx	m	−0.0104	0.0044	5.48	0.0193	0.95	0.91–0.99
Nasopharynx	f	−0.0024	0.0084	0.08	0.7791	0.99	0.91–1.07
Hypopharynx	m	−0.0156	0.0049	10.15	0.0014	0.92	0.88–0.97
Hypopharynx	f	−0.0053	0.0091	0.34	0.5592	0.97	0.89–1.06
Larynx	m	0.0009	0.0038	0.06	0.8088	1.00	0.97–1.04
Larynx	f	0.0127	0.0060	4.51	0.0337	1.07	1.00–1.13

m = male; f = female; StdErr = standard error; ChiSq = chi-square distribution; CI = confidence interval.

## Data Availability

The datasets used during the current study are available from the corresponding author upon reasonable request.

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
