# Peer review of "Trends in Treatment of Head and Neck Cancer in Germany: A Diagnosis-Related-Groups-Based Nationwide Analysis, 2005–2018"

_cancers, 2021, doi:10.3390/cancers13236060_

Round 1
Reviewer 1 Report
In this paper the Authors used DRG data from 2005 to 2018 to analyze nationwide treatment rates for head and neck cancers in Germany.
I have two comments:
- in the paper the Authors use the voice "chemotherapy/ immunotherapy". What is meant by immunotherapy? Since checkpoint inhibitors did not play a role in the examined period?
- in the discussion there are few references that support the results. In my opinion it would be interesting to expand the discussion to have a more critical view of the results.
Author Response
Thank you for the detailed and helpful review.
We answered all queries/comments of the reviewer.

Reviewer 2 Report
Hermanns et al. describe the inpatient treatment type of HNSCC patients based on ICD-10 (German modification) and ICPM (German modification) data, which are compulsorily collected from all inpatient hospital cases in the context of hospital accounting by the DRG system. Statistically, a GLM with Poisson or Negative Binomial distribution was correctly used and modeled in SAS. The introduction and discussion are well written. The results are a bit unfocused. Material and methods is adequate. Overall, the manuscript is good and interesting, so I would like to congratulate the authors for this good manuscript. Therefore, as an oral and maxillofacial surgeon and a medical informatics specialist, I was very interested in several aspects of the manuscript. The manuscript is therefore basically worthy of publication.
In my eyes, too many tables and graphs are presented. I recommend the authors to put all actual tables and graphs in the Supplementary. Instead, the most important findings should be presented as graphs and tables in such a way that they are immediately apparent to the reader. The German saying "Man sieht vor lauter Bäumen den Wald nicht mehr" (You can't see the forest for the trees) sums it up quite well. It is important to note that the reader group is primarily clinicians, so this should be kept in mind respectively.
The authors write something about the number of average procedures per year under 3.1. "Treatment strategies and average treatment rates". The mean values seem to be wrong by a factor of 10. Did the authors make a mistake here? If the mean of biopsies is 2600 per year, how can a total of 218000 biopsies have been performed in 14 years?
Because of the many statistical models, a correction for multiple testing should be done.
What confuses me a bit is that neck dissection does not correlate strongly with surgery. Actually, according to the guideline, when tumor is removed in the oral cavity, neck dissection is also usually performed. How do the authors explain this?
In Figure 1 and 2, for example, percentage changes to a base (mean) could be shown. One could also draw a horizontal line here that corresponds to the mean value. The Y-axis can also be displayed logarithmically or pseudo logarithmically. In some cases, it is not possible to distinguish between the different curves.
In table 1, women and men should be displayed next to each other, i.e. horizontally. Standard deviations I would always indicate with mean±SD, instead of separate columns. I would define the localization as a column, so simply rotate the table by 90°.
Table 2 I would also rotate by 90° and use the symbols + (increase), - (decrease) and Ø (unchanged). I would also arrange women and men next to each other so that they can be easily compared.
In table 3-7 I would put the confidence intervals in brackets, i.e. RR (X - Y). You should also consider whether it is really necessary to show so many decimal places
Author Response

(The authors gave the same response as above.)
